# Advances in the Molecular Cytogenetics of Bananas, Family Musaceae

**DOI:** 10.3390/plants11040482

**Published:** 2022-02-11

**Authors:** Denisa Šimoníková, Jana Čížková, Veronika Zoulová, Pavla Christelová, Eva Hřibová

**Affiliations:** 1Institute of Experimental Botany of the Czech Academy of Sciences, Centre of the Region Hana for Biotechnological and Agricultural Research, 77900 Olomouc, Czech Republic; simonikova@ueb.cas.cz (D.Š.); cizkova@ueb.cas.cz (J.Č.); veronika.zoulova01@upol.cz (V.Z.); christelova@ueb.cas.cz (P.C.); 2Department of Cell Biology and Genetics, Faculty of Science, Palacký University, 77900 Olomouc, Czech Republic

**Keywords:** flow cytometry, chromosomes, fluorescence in situ hybridization, rRNA genes, DNA repeats, BAC clones, oligo painting, karyotyping

## Abstract

The banana is a staple food crop and represents an important trade commodity for millions of people living in tropical and subtropical countries. The most important edible banana clones originated from natural crosses between diploid *Musa balbisiana* and various subspecies of *M. acuminata*. It is worth mentioning that evolution and speciation in the Musaceae family were accompanied by large-scale chromosome structural changes, indicating possible reasons for lower fertility or complete sterility of these vegetatively propagated clones. Chromosomal changes, often accompanied by changes in genome size, are one of the driving forces underlying speciation in plants. They can clarify the genomic constitution of edible bananas and shed light on their origin and on diversification processes in members of the Musaceae family. This article reviews the development of molecular cytogenetic approaches, ranging from classical fluorescence in situ hybridization (FISH) using common cytogenetic markers to oligo painting FISH. We discuss differences in genome size and chromosome number across the Musaceae family in addition to the development of new chromosome-specific cytogenetic probes and their use in genome structure and comparative karyotype analysis. The impact of these methodological advances on our knowledge of *Musa* genome evolution at the chromosomal level is demonstrated. In addition to citing published results, we include our own new unpublished results and outline future applications of molecular cytogenetics in banana research.

## 1. Introduction

Bananas (*Musa* ssp.) are large herbaceous plants grown in tropical and subtropical regions of Southeast Asia, Africa, and South America [1,2]. The banana is one of the most important crops cultivated worldwide, with annual production exceeding 150 million tons [3]. Cooking bananas serve as a staple food for millions of people. Moreover, bananas are one of the major export commodities of several developing countries, establishing them as an important element of national trades with a significant socioeconomic role.

The genus *Musa*, together with the two closely related genera *Ensete* and *Musella*, is a member of the monocotyledonous family Musaceae. Whereas *Ensete* and *Musella* are represented by only a few species, the genus *Musa* comprises about 75 species and numerous cultivated edible clones. Wild *Musa* species have traditionally been subdivided into four sections based on the basic chromosome number (x) and a set of morphological descriptors [4,5]. These sections, namely Eumusa (x = 11), Rhodochlamys (x = 11), Australimusa (x = 10), and Callimusa (x = 9, 10), were later supplemented by a fifth section Ingentimusa, which contains a single species *Musa ingens* (x = 7) [6].

Modern cultivated bananas originated from natural inter- and intraspecific crosses between two wild diploid species of the Eumusa section: *M. acuminata* (A genome) and *M. balbisiana* (B genome). The evolution of some cultivars also involved crosses with *M. schizocarpa* (S genome) and Australimusa species (*M. textilis*, T genome), as revealed by molecular and cytogenetic methods [7,8,9,10]. A unique group of edible bananas is represented by Fe’i bananas found in the Pacific. These bananas are believed to originate from ancestors belonging to the Australimusa section [11,12]. However, most edible bananas are vegetatively propagated diploid (AA, AB), triploid (AAA, AAB, or ABB), or tetraploid clones classified in groups based on the relative proportion of respective *M. acuminata* and *M. balbisiana* genomes in their genotypes [13].

Cultivated bananas’ complex mode of origin and the insufficient resolution of morpho-taxonomic classification markers call for more detailed studies of the *Musa* genome on chromosomal and DNA levels. Moreover, information about chromosome structural changes could help banana breeders select appropriate parents during banana breeding processes. Traditional cytogenetic methods, such as for estimating genome size, ploidy, and chromosome counting, provide important information about the genome. Additional information on genomic constitution and large-scale chromosome structural changes that are associated with the evolution and speciation of bananas and their closely related genera can be obtained using molecular cytogenetics methods such as fluorescence in situ hybridization (FISH) and genomic in situ hybridization (GISH). 

## 2. Nuclear Genome of Bananas

Nuclear genome size is one of the basic characteristics of a species and is commonly used in taxonomic studies of higher plants [14,15]. This value represents one copy of nuclear genetic information (equal to 1C) and can be expressed either in picograms (pg) or gigabase pairs (Gbp) using the formula of Doležel et al. [16], which considers 1 pg DNA to be equal to 0.978 × 109 bp. A genome size of ~0.6–0.64 pg/1C was estimated for *M. acuminata* and ~ 0.55–0.57 pg/1C for *M. balbisiana* using flow cytometry, which clearly discriminated both species [8,17,18]. D’Hont et al. [19], Bartoš et al. [20], and Čížková et al. [21] further extended the knowledge on Musaceae nuclear DNA content by analyzing wild *Musa* and *Ensete* species. Their studies showed that the genome sizes of Eumusa and Rhodochlamys accessions overlap, ranging from 0.61 to 0.69 pg/1C in Eumusa and from 0.6 to 0.66 pg/1C in Rhodochlamys. Australimusa species were characterized by genome sizes varying between 0.65 and 0.77 pg/1C. The highest nuclear DNA content and interspecific variation of this feature were observed in species belonging to the Callimusa section, with values ranging from 0.7 to 0.89 pg/1C. A strong negative correlation between the 2n chromosome number and nuclear DNA content was found within *Musa* species [21].

The genome size was also estimated for *Ensete gilletii* [20] and *Musella lasiocarpa* (Figure 1A, unpublished) to complement our knowledge of nuclear genomes in all three genera. Representatives of two genera closely related to genus *Musa* had nuclear DNA contents similar to the Eumusa and Rhodochlamys species. *E. gilletii* was characterized by a genome size of 0.61 pg/1C [20], and an even lower value (0.59 pg/1C) was estimated for *M. lasiocarpa* (Figure 1A, unpublished).

The low level of divergence in genome sizes within species of the Eumusa section containing diploid, triploid, and tetraploid edible banana cultivars permit the use of flow cytometric analysis for rapid identification of ploidy levels. This approach is based on nuclei isolation from a small part of banana leaf tissue used in combination with an external or internal standard. In the case of bananas, previously characterized diploid *M. acuminata* or *M. balbisiana* genotypes can serve as standards for ploidy estimation of economically important intra- and interspecific banana hybrid clones that originated from natural crosses between these two diploid species or during banana breeding programs. As shown in the study by Christelová et al. [9], ploidy level estimation using flow cytometry can be assessed using chicken red blood cell nuclei (CRBC) as an internal reference standard. The ratio between the relative DAPI fluorescence intensity of G1-phase nuclei from accessions of the *Musa* section and CRBC nuclei is ~0.5 for diploid plants, ~0.75 for triploid plants, and 1 for tetraploid plants (Figure 2, unpublished). Aneuploidy may be the case for plants with ploidy levels deviating from regular ratios. Roux et al. [22] successfully applied flow cytometry to detect aneuploidy in bananas. On the other hand, their study also revealed that this method is laborious and inconvenient for large-scale screening.

## 3. Chromosome Number and Structure 

A small genome of bananas (genus *Musa*) and related genera are divided into a relatively high number of morphologically similar chromosomes. The basic chromosome number (x) in species of the Musaceae family varies from 7 to 11 (Figure 3). Closely related genera *Ensete* and *Musella* are diploid with a basic chromosome number x = 9. Basic chromosome number and plant morphology are traditionally used characteristics for taxonomically classifying members of the *Musa* genus, which is divided into four sections: Eumusa (x = 11), Rhodochlamys (x = 11), Australimusa (x = 10), and Callimusa (x = 7, 9, 10) [4] (Figure 3). *Musa ingens* , assigned to the Callimusa section, represent a single species with the lowest basic chromosome number (x = 7). Most edible banana clones originated as a result of natural intra- and interspecific hybridization between subspecies of *M. acuminata* (A genome) and *M. balbisiana* (B genome) within section Eumusa and contain 22 (2n = 2x = 22, diploids), 33 (2n = 3x = 33, triploids), and 44 (2n = 4x = 44, tetraploids) chromosomes in somatic cells. Another small but nutritionally valued group of edible banana clones, known as Fe’i bananas, is derived from *M. maclayi*, a member of the Australimusa section. They are diploids and have maintained the basic chromosome number characteristic of representatives of the Australimusa section (x = 10) [1,20,23]. 

Banana mitotic chromosomes are usually 1–2 μm long meta- or submetacentric and have similar morphology (Figure 4A–D). Differences in chromosome morphology were observed in species of the genera *Ensete* and *Musella* and section *Callimusa.* Although some of their mitotic metaphase chromosomes are larger (1–4 μm), it is impossible to unambiguously identify all individual chromosomes using standard chromosome features, such as chromosome length and position of centromere and secondary constriction(s) [8,20,21,25,26,27]. In addition to the small size of mitotic metaphase chromosomes, the high level of chromatin condensation also complicates their identification. Similarly, secondary constriction does not serve as a good morphological marker for chromosome identification in *Musa*. Although secondary constrictions are easily localized in *Musa* species (Figure 4A–D) [25], they readily break away from the rest of the chromosome during chromosome spread preparation (Figure 4D). In more recent studies, secondary constrictions separated from mother chromosomes were considered supernumerary small chromosomes [28,29]. In analyzed accessions within the sections Eumusa, Rhodochlamys, Australimusa, and Callimusa, the number of secondary constrictions corresponds to their ploidy levels (Figure 4A–D) [8,25,26]. A higher variability (higher number) of secondary constrictions was observed in species from the genera *Ensete* [20] and *Musella* (Figure 1B, unpublished). 

The development of chromosome banding techniques in the 1970s and 1980s [30,31,32] was effective for chromosome identification in humans and some animals. Unfortunately, banding techniques were successful only in the case of a few plant species (e.g., species of the family Poaceae [33,34,35] or *Lilium* [36]), which are commonly characterized by large genomes containing numerous repeat-rich regions. To identify individual chromosomes in most plant species, including those of Musaceae, molecular cytogenetics techniques that utilize specific chromosome landmarks localized by in situ hybridization must be applied. 

## 4. Molecular Cytogenetics of Musaceae 

The development of in situ hybridization and, particularly, fluorescence in situ hybridization (FISH) [37] has enabled the detailed analysis of genome and chromosome structure. FISH is a suitable method for localizing fluorescently labeled DNA sequences (probes) onto mitotic or meiotic chromosomes or in the nuclei and provides important insights into the long-range organization of DNA sequences in the genomes [37,38]. In many plant species, including bananas, FISH presents the only possibility for discriminating individual chromosomes and generating molecular karyotypes. The potential of FISH for karyotyping depends on the availability of probes that produce chromosome-specific labeling patterns.

Typical chromosomal loci largely used as FISH probes are represented by tandem repetitive arrays of 5S and 45S rRNA genes and satellite DNA sequences whose chromosome location is species-specific (e.g., [25,39,40,41,42]). Another extensively employed probe types are BAC (bacterial artificial chromosome) clones bearing large single- and low-copy DNA sequences [43,44] or certain mobile repetitive elements dispersed on all chromosomes but provide signals on specific chromosomal regions, e.g., in centromeres [8,45,46,47]. Recently, short single-copy sequences, mainly derived from genic regions (1.5–5 kb long), were also successfully used for karyotyping in some plant species, e.g., wheat, barley, and *Agropyron* spp. [48,49,50]. However, short single-copy sequences were not used as probes for FISH due to the high level of mitotic *Musa* chromosome condensation. 

### 4.1. Identification of Chromosomes by FISH

Even though species of the Musaceae family have small genomes, only a few DNA sequences have proved useful in studying the organization of nuclear genomes at the cytogenetic level to date (Figure 5A–K). In the first studies, probes specific to 5S and 45S rDNA sequences were used to identify mitotic chromosomes of progenitors of the most popular edible banana clones-*M. acuminata* and *M. balbisiana* [25,26]. Genes for 45S rRNA were localized onto a secondary constriction (nucleolar organizing region, NOR) for one pair of chromosomes in both species, and differences were found in the 5S rDNA loci between *M. acuminata* and *M. balbisiana*. Since 5S and 45S rRNA genes occur in thousands of tandemly organized copies, usually localized in separate loci within the genomes, they serve as popular cytogenetic landmarks in many plant species and are still widely used, e.g., in *Aegilops* spp. [51], *Brassica* spp. [52] or *Fragaria* spp. [53]. The results from further cytogenetic studies of Musaceae have contributed to our knowledge on the organization of rRNA genes in the genomes of more than 60 accessions, including various *Musa* species, edible banana cultivars, and of the related genera *Ensete* and *Musella* [8,10,20,21,25,26,54]. 

Genes coding for a large rRNA subunit (45S) were localized to the NOR region on one chromosome pair of diploid, three chromosomes of triploid, and four chromosomes of tetraploid representatives in the Eumusa section (Figure 5H) [8,20,25,26]. Species of the closely related Rhodochlamys section contain one or two pairs of mitotic metaphase chromosomes with 45S rDNA signals localized in secondary constrictions; representatives of the Australimusa section contained one chromosome pair carrying NOR [20,21]. The situation within the Calimusa section is more variable, reflecting the high genetic species diversity in which there are even differences in the number of chromosomes [20,21]. Most Callimusa species have 20 mitotic chromosomes with two 45S rDNA loci localized in secondary constrictions. On the other hand, *M. borneensis* (2n = 2x = 20) was found to contain five 45S rDNA loci, with two localized in secondary constrictions; however, the other two signals were large interstitial clusters on other chromosomes. The fifth weak signal was detected in the telocentric region of another chromosome [21]. A similar distribution of 45S rDNA loci was also observed in *M. beccarii* with 2n = 2x = 18, where two loci were located in secondary constrictions and another four strong interstitial clusters were located on other chromosomes [20,21]. Despite having the same basic number of chromosomes (2n = 2x = 18), the phylogenetically related genus *Ensete* even contains eight 45S rDNA loci in secondary constrictions located at the chromosomal ends, whereas *Musella* genus contains four loci of 45S rRNA genes (Figure 1B, unpublished).

A larger variation was observed in 5S rDNA loci, localized on chromosomes other than those carrying 45S rRNA genes in NOR (Figure 5H) [8,10,20,21,25,26,54]. The only exception was observed in *M. beccarii* and *M. borneensis*, in which two of the 5S rDNA loci co-localized with interstitially localized 45S rRNA genes [21]. In contrast to most 45S rDNA location, odd numbers of 5S rDNA loci were observed in more *Musa* representatives, e.g., in *M. acuminata* ‘Pisang Mas’, ‘Pisang Lilin’, ‘Niyarma Yik’, and the hybrid ‘TMPx 8084-1′ [20,21,25,26].

Given the lower chromosome number, one may speculate that the interstitial clusters of 45S rDNA are a remnant of chromosome reshuffling accompanying evolution. The odd number of 45S and 5S rDNA loci could indicate structural chromosome heterozygosity, suggesting the hybrid origin of some *Musa* species. In this context, the presence of odd numbers of 5S rDNA in interspecific hybrid clones between *M. acuminata* and *M. schizocarpa* [21] may be interpreted as evidence for such a hypothesis. On the other hand, unstable additional chromosome fragments may cause variability in rDNA loci, as shown in banana hybrid clones, e.g., TMPx 8084-1 (AA), obtained from a cross between wild diploid genotype *M. acuminata* ‘Calcutta 4′ and a triploid plantain [26]. However, the 5S rDNA loci contained a low number of repetitive units, implying that hybridization signals below the detection limit of FISH cannot be overlooked as a causal agent for the odd number of 5S rDNA loci [21].

Given the high basic chromosome number of most Musaceae representatives, co-localization of 5S and 45S rDNA loci did not permit chromosome identification. Therefore, other types of repetitive DNA sequences and BAC clones were employed to generate molecular karyotypes in Musaceae. Former studies focused on isolating and characterizing different types of repetitive DNA sequences, mainly tandem organized repeats, which serve as good cytogenetic markers in plant studies, e.g., in wheat [55,56], *Aegilops* spp. [57], or *Agropyron cristatum* [50]. Balint–Kurti et al. [58] as well as Valárik et al. [27] and Hřibová et al. [59] succeeded in identifying one sequence region of retrotransposon monkey and repetitive elements named Radka1 and Radka 7, both of which localized exclusively to NOR and co-localized with the 45S rDNA loci. Unfortunately, further analysis of the repetitive parts of banana and enset genomes by low coverage next-gen sequencing [60] provided only two candidates for tandem repeats that performed well as cytogenetic markers. A more complex analysis of the banana repeatome performed in five *Musa* and one *Ensete* species [60,61] confirmed the two main satellite DNA sequences, and these were later successfully used for karyotyping in *Musa* (Figure 5) [8,10,54]. However, a more detailed characterization of the repeats led to identification of a retrotransposon MusA1 and a LINE element, predominantly localized in the centromeric region and further successfully used as centromeric probes in *Musa* cytogenetic studies (Figure 5D) [8,10,62].

The traditionally employed chromosome-specific set of landmarks also covers single- and low-copy BAC (bacterial artificial chromosome) clones, which were used to identify specific chromosomes in several plant species [63,64,65,66]. The BAC-FISH was largely unsuccessful in bananas when most low-copy BAC clones identified after a colony array of BAC libraries provided dispersed signals in the genome, even after their subcloning [46]. Only a few BAC clones were mapped onto banana mitotic and meiotic chromosomes as chromosome specific (Figure 5G) [8,46,67,68]. However, new BAC clones selected from BamH1 and HindIII BAC libraries of *M. acuminata* ssp. *malaccensis* ‘DH Pahang’ [69] were used to validate chromosomal translocations detected in silico [70,71].

FISH probes derived from BAC clones or repetitive DNA sequences have been widely applied in many species; however, preparation is time consuming. Moreover, these probes are inefficient in plant species with many chromosomes or when the chromosome size is an issue [38,45]. Altogether, standard cytogenetic landmarks, including satellite DNA and other repetitive DNA sequences or single- and low-copy BAC clones, did not result in the unambiguous identification of all banana chromosomes at the cytogenetic level.

### 4.2. Integrated Karyotyping Using Oligo Painting FISH

The possibility to design probes capable of distinguishing between individual chromosomes and comparative karyotype analysis in plant species was recently achieved using oligo painting FISH. This method is based on the identification of large sets of unique oligomer sequences (up to 50 nt long) specific to selected genome regions, e.g., chromosomes (reviewed in Jiang [72]).

Chromosome-scale assemblies or at least draft genome assemblies are used for in silico identification of all possible unique oligomers that can be used as probes for FISH (e.g., [73,74]). Computational analysis and in silico selection of oligomers covering specific genome regions can be performed using Chorus software, specifically created for this approach [75] or other bioinformatic tools (reviewed in Liu and Zhang [76]). Large sets of oligomers are then synthesized in parallel, labeled directly by fluorophores or indirectly by haptens, and used for in situ hybridization. Oligo painting FISH has successfully been used to analyze the genome structure in many plant species, including Poaceae species [77,78,79,80,81,82,83,84], *Solanum* spp. [85,86] or subtribe Phaseolineae [87].

In bananas, oligo painting FISH was established by Šimoníková et al. [54], who created chromosome arm-specific painting probes based on the reference genome sequence of doubled haploid *M. acuminata* ssp. *malaccensis* ‘DH Pahang’ v.2 (A genome, x = 11) [88]. In their study, a set of nineteen oligo painting probes was successfully hybridized to mitotic chromosomes of *M. acuminata*, and two closely related species, *M. balbisiana* (B genome, x = 11) and *M. schizocarpa* (S genome, x = 11), that participated in the origin of most edible banana clones. For the first time in *Musa* cytogenetics, all individual chromosomes were unambiguously distinguished, and molecular karyotypes of the three *Musa* species were established by combining oligo painting probes and previously described cytogenetic markers [54]. Oligo painting probes facilitated the anchoring of pseudomolecules to chromosomes and identifying large translocation events in *M. balbisiana*. Moreover, the density of these oligo painting probes was sufficient for analyzing less condensed meiotic pachytene chromosomes (Figure 6A,B), thereby opening another important future application for detailed analysis (or confirmation) of chromosomal rearrangements detected in silico, e.g., in draft genome assemblies created by long-read sequencing technologies.

Afterward, the chromosome-arm specific oligo painting probes were applied to study genome structure in twenty representatives of genus *Musa*, covering edible banana clones and their probable progenitors [89]. This method was used to detect many chromosomal translocations that were specific to individual subspecies of *M. acuminata.* Interestingly, an odd number of translocation events was noticed in the genomes of some analyzed wild diploid *Musa* species (e.g., *M. acuminata* ssp. *siamea,* and ssp. *bu**rmannica*), indicating their hybrid origin and the structural heterozygosity of their genomes. Therefore, oligo painting FISH was shown to be useful for identifying edible banana clone progenitors and brought new insights into their evolution. As an example, oligo painting FISH revealed a reciprocal translocation between chromosomes 3 and 8 in two chromosome sets in triploid East African highland banana (EAHB) clone ‘Imbogo’ (Figure 7C,D). The same translocation was detected in the homozygous state of *M. acuminata* ssp. *zebrina ‘*Maia Oa’ (Figure 7A,B). Thus, *M. acuminata* ssp. *zebrina* has been acknowledged as one of the progenitors of East African highland bananas [89].

### 4.3. Analysis of Genome Constitution in Banana Hybrid Clones

Many plant species and economically important cultivars, including edible banana clones, have originated from intra- or interspecific crosses. Thus, one of the important tasks is to discover the genomic constitution of interspecific hybrid species by identifying parental chromosomes and their recombination products at the cytogenetic level, which can be achieved by applying genomic in situ hybridization (GISH). Compared to FISH, where specific DNA sequences localize to genomes, GISH uses total genomic DNA to probe chromosomes. GISH was successfully used to analyze chromosome constitution and meiotic pairing in different natural or artificial plant hybrids and even allopolyploids, including representatives of the Poaceae, Brassicaceae, and Solanaceae families [90,91,92,93,94,95]. During the GISH procedure, the total genomic DNA of one (putative) progenitor serves as a probe, and genomic DNA isolated from other (putative) progenitors of the studied hybrid or allopolyploid specimen serve either as blocking DNA, additionally used in excess amounts relative to the probe, or as a second-color probe. The probe/blocking DNA ratio is species-specific and depends on the degree of homology between the progenitors of the hybrid or allopolyploid species [90,96,97]. 

The same applies to banana hybrid clones, where the success of GISH experiments is highly dependent on phylogenetic relatedness and the variability of the genomes that gave rise to the hybrids. The most important interspecific banana cultivars originated by crosses between two closely related wild species *M. acuminata* (A genome) and *M. balbisiana* (B genome), both representatives of the Eumusa section (Figure 3). GISH experiments performed on A × B banana hybrids resulted in signals covering all chromosomes (with different intensities) in the hybrid genomes. Moreover, the signals covered only pericentromeric regions and partially distal chromosome parts [97,98,99]. The labeling pattern in A × B banana hybrids did not visualize distal and subtelomeric chromosome parts often involved in translocation events. Thus, it was also impossible to detect recombination events between interspecific chromosomes [97,98,99]. These findings demonstrate the limits of the GISH technique for analyzing the genomic constitution of A × B banana hybrid clones. 

By comparison, A × T banana hybrid clones originate from crosses between distantly related species *M. acuminata* (A genome, Eumusa section) and *M. textilis* (T genome, Australimusa section). In these banana hybrids, GISH clearly labeled chromosomes specific to A and T subgenomes [97], suggesting the presence of more differentiated DNA repeat types between these two species [61]. FISH experiments were conducted with the two Radka DNA sequences (Radka5 and Radka6) [27] using genomes of A × T banana hybrid clones to demonstrate the main contribution of specific DNA repeats to GISH signals. Radka DNA sequences were initially isolated from *M. acuminata* and cytogenetically localized to all mitotic chromosomes in *M. acuminata* and *M. balbisiana* [27]. These two short parts (742, resp. 193 nt long) of banana retroelements were determined to be specific to Eumusa species, and FISH analysis with probes derived from Radka5 and Radka6 in the genome of A × T banana hybrids resulted in signals specific (visible) to chromosomes originating from the A subgenome progenitor (Figure 8A,B, unpublished). Compared with A × B banana hybrid clones, whose progenitors are more closely related and share a large proportion of similar DNA sequences, progenitors of A × T banana hybrid clones are more phylogenetically distant. Only a short part of the repetitive elements was sufficient for visualizing chromosomes of one particular progenitor by in situ hybridization. 

## 5. Conclusions and Future Perspectives

Current cytogenetic studies on bananas benefit from combining the advantages of two complementary techniques, flow cytometry and fluorescence in situ hybridization (FISH). Flow cytometry is a rapid and reliable method for estimating genome size and ploidy level within the whole Musaceae family. However, this technique cannot be used to determine the chromosome number or detect potential aneuploidy. Methods of molecular cytogenetics based on fluorescence in situ hybridization (FISH) are essential for the unambiguous identification of chromosomes and their number in the genome. Moreover, FISH provides valuable information on genome structural changes that accompany speciation processes.

Traditional probes used for FISH, such as various tandem or dispersed repetitive DNA sequences, rRNA genes, and a few BAC clones, could not identify the individual chromosomes of the three Musaceae genera and did not facilitate comparative karyotype analysis. Oligo painting FISH represents a breakthrough technique in banana molecular cytogenetics. For the first time, all *Musa* chromosomes were individually identified along with the variation in chromosome structure within different *Musa* species. This knowledge helped in reconstructing their molecular karyotypes and conducting comparative cytogenetic studies that led to the formation of an independent theory regarding additional evolutionary mechanisms underlying banana speciation. Moreover, information about chromosome structural changes may help banana breeders select appropriate parents during banana breeding processes more efficiently.

Furthermore, chromosome-specific oligo painting was used to analyze more decondensed meiotic pachytene chromosomes and represents a promising tool for the in situ analysis of chromosome pairing during meiosis, especially in interspecific banana hybrid clones consisting of subgenomes varying in unbalanced translocations. As a complementary approach to genomic in situ hybridization (GISH) used to distinguish parental genomes in interspecific banana hybrids, species- or subspecies-specific DNA repeats can be used to unambiguously discriminate parental chromosomes of interspecific hybrids. Apart from their use in the GISH technique which did not ensure successful visualization of entire chromosomes, especially in the most important banana hybrid clones originating from crosses between two closely related species (*M. acuminata* and *M. balbisiana*), their application in the identification of species-specific repeats as cytogenetic markers can be highly beneficial. Finally, chromosome arm-specific oligo painting probes developed for banana can be used to analyze the three-dimensional organization and positioning of chromosomes during interphase in detail.

## Figures and Tables

**Figure 1 plants-11-00482-f001:**
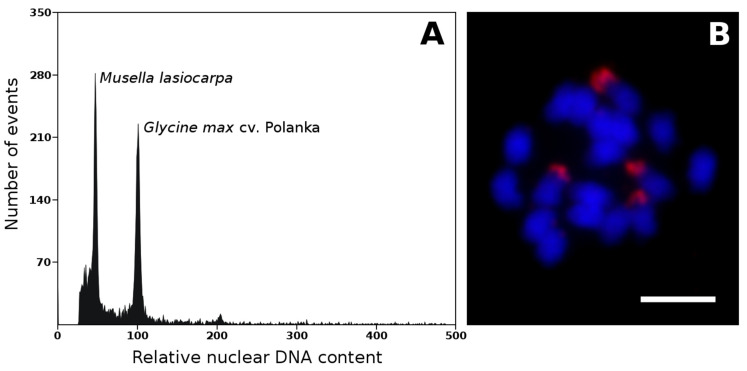
Estimation of nuclear genome size and chromosome number in *Musella lasiocarpa*. (**A**) Histogram of relative nuclear DNA content obtained after flow cytometric analysis of propidium iodide-stained G1 nuclei of *M. lasiocarpa* (2C = 1.174 pg) and *Glycine max* (2C = 2.5 pg), which served as the internal reference standard. Measurements were performed according to Bartoš et al. [20]. (**B**) mitotic metaphase chromosomes of *M. lasiocarpa* (2n = 2x = 18). The chromosomes were counterstained with DAPI (blue); the probe specific to 45S rDNA sequence (red) was localized to secondary constrictions. Probe specific to 45S rDNA was prepared according to Čížková et al. [8]. Bars correspond to 5 µm.

**Figure 2 plants-11-00482-f002:**
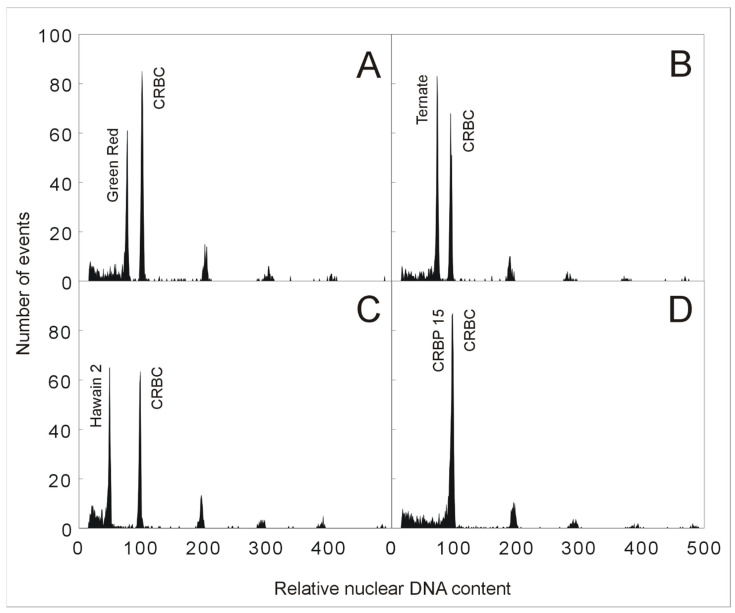
Determination of *Musa* ploidy level by flow cytometry. Histograms of relative nuclear DNA content obtained after flow cytometric analysis of DAPI-stained nuclei isolated from various accessions of *Musa*, and chicken red blood cell nuclei (CRBC) served as the internal reference standard. (**A**) triploid accession ‘Green Red’ (genomic constitution AAA; peak ratio 0.75); (**B**) triploid accession ‘Ternate’ (genomic constitution AAB; peak ratio 0.76); (**C**) diploid *M. acuminata* ‘Hawain 2′ (genomic constitution AA; peak ratio 0.51); (**D**) tetraploid accession ‘CRBP 15′ (genomic constitution AAAB; peak ratio 1). Measurements were performed according to Christelová et al. [9].

**Figure 3 plants-11-00482-f003:**
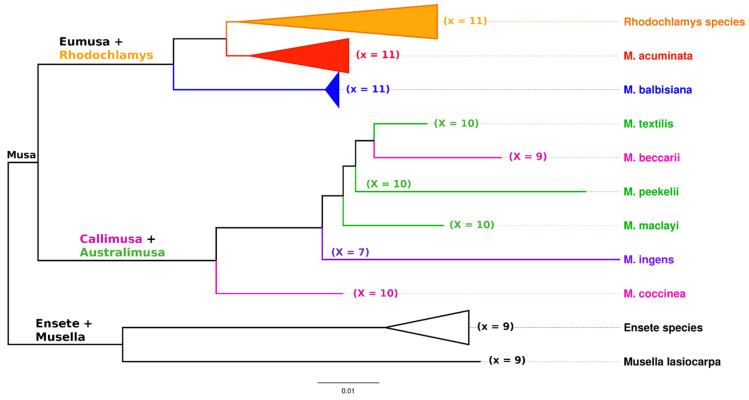
Phylogenetic tree of Musaceae family members. The tree is constructed from ITS1–ITS2 sequence regions of ribosomal DNA using the BioNJ method implemented in SeaView [24]. Names of the four sections of *Musa*, species belonging to the corresponding section, and their basic chromosome number (x) are printed in different colors: Eumusa section (red), resp. (blue), Rhodochlamys section (orange), Callimusa section (pink), and Australimusa section (green). In addition, *M. ingens* from the Ingentimusa section is included in the figure (purple).

**Figure 4 plants-11-00482-f004:**
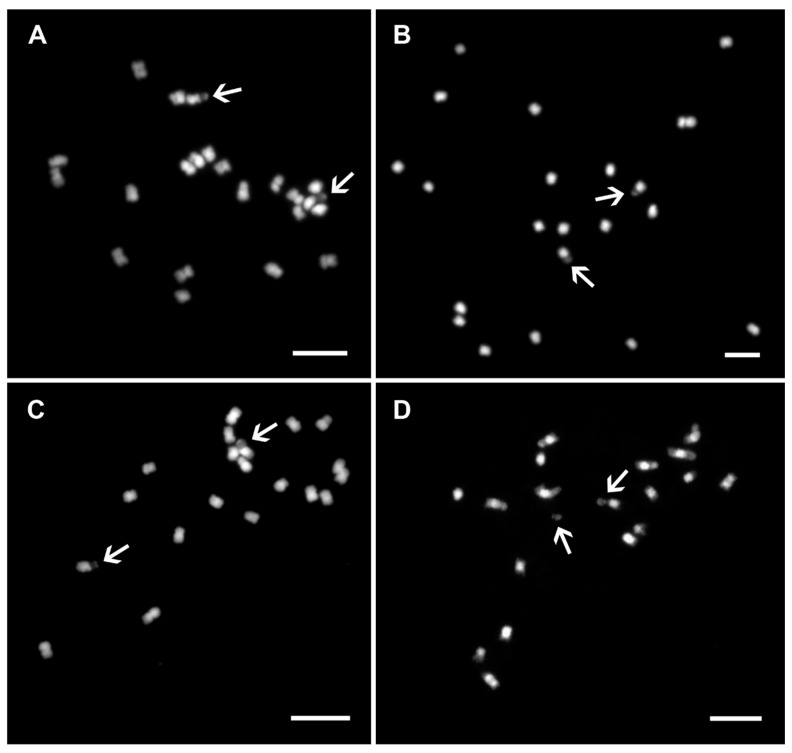
Mitotic chromosomes of genus *Musa*. Mitotic metaphase plates of (**A**) *Musa acuminata* ssp. *burmannicoides* ITC0249 (2n = 2x = 22) from Eumusa section; (**B**) *M. laterita* ITC1575 (2n = 2x = 22) from Rhodochlamys section; (**C**) *M. maclayi* ITC1207 (2n = 2x = 20) from Australimusa section; and (**D**) *M. beccarii* ITC1070 (2n = 2x = 18) from Callimusa section. The chromosomes were stained with DAPI, and the images are shown in white pseudo-color. Arrows indicate secondary constrictions. One secondary constriction broke from the rest of the chromosome as shown in (**D**). Bars correspond to 5 µm.

**Figure 5 plants-11-00482-f005:**
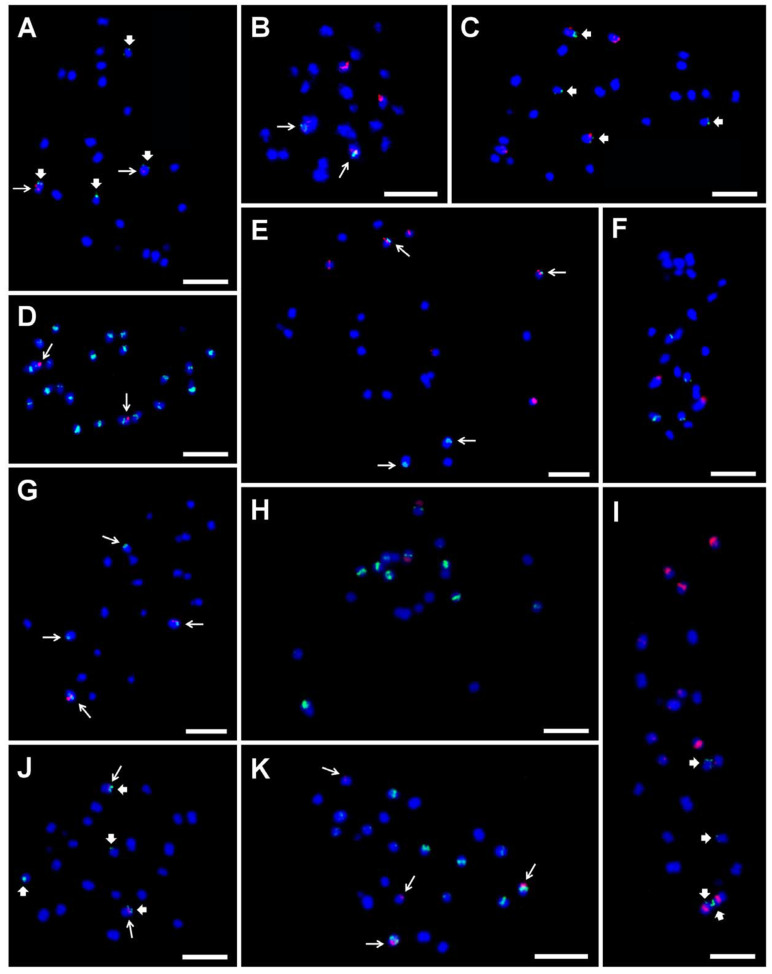
Examples of the genomic distribution of DNA satellites, BAC clone, LINE element, and rDNA as determined in mitotic metaphase chromosomes of diploid *Musa* accessions after FISH. Satellite repeat sites CL18 and CL33 probe hybridization are marked by long and thick arrows, respectively. (**A**) CL18 (red) and CL33 (green) on ‘Long Tavoy’ chromosomes; (**B**) 5S rDNA (red) and CL18 (green) on ‘Maia Oa’ chromosomes; (**C**) 5S rDNA (red) and CL33 (green) ‘Long Tavoy’ chromosomes; (**D**) CL18 (red) and LINE element (green) on ‘Tuu Gia’ chromosomes; (**E**) 5S rDNA (red) and CL18 (green) on ‘Pisang Klutuk Wulung’ chromosomes; (**F**) BAC clone 2G17 (red) and 5S rDNA (green) on chromosomes of ‘Cameroun’; (**G**) BAC clone 2G17 (red) and CL18 (green) on ‘Tani’ chromosomes; (**H**) 45S rDNA (red) and 5S rDNA (green) on *M. schizocarpa* ITC 0560 chromosomes; (**I**) 5S rDNA (red) and CL33 (green) on *M. schizocarpa* ITC 1002 chromosomes; (**J**) CL18 (red) and CL33 (green) on *M. schizocarpa* ITC 1002 chromosomes; (**K**) CL18 (red) and 5S rDNA (green) on *M. schizocarpa* ITC 1002 chromosomes. Chromosomes were counterstained with DAPI (blue). Bars correspond to 5 µm. Retrieved from Čížková et al. [8].

**Figure 6 plants-11-00482-f006:**
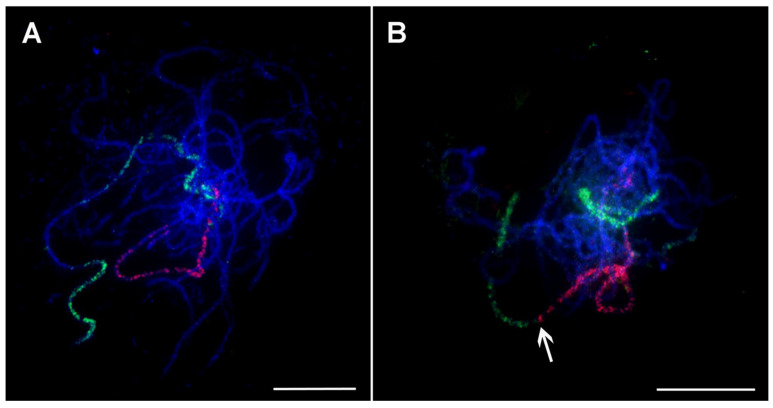
Oligo painting FISH on meiotic pachytene chromosome spreads of *Musa*. (**A**) *M. acuminata* ssp. *malaccensis* ‘Pahang’ (2n = 2x = 22, AA; chromosome 1 in red, chromosome 4 in green); (**B**) *M. balbisiana* ‘Tani’ (2n = 2x = 22, BB; chromosome 1 in red, chromosome 3 in green). Chromosomes were counterstained with DAPI (blue). Arrows point to the region translocated from chromosome 3 to chromosome 1 in *M. balbisiana*. Bars correspond to 10 µm. Retrieved from Šimoníková et al. [54].

**Figure 7 plants-11-00482-f007:**
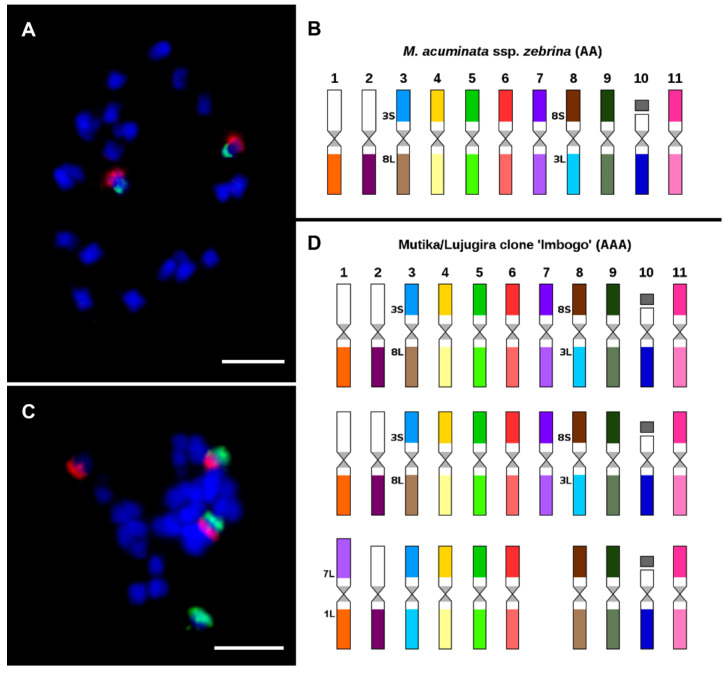
Oligo painting FISH on mitotic metaphase plates of two *Musa* accessions (**A**) *M. acuminata* ssp. *zebrina* ‘Maia Oa’ (2n = 2x = 22, AA; long arm of chromosome 3 in red, short arm of chromosome 8S green; (**B**) aneuploid East African Highland banana (EAHB) clone ‘Imbogo’ (2n = 3x − 1 = 32, AAA; long arm of chromosome 3 in green, short arm of chromosome 8 in red; (**C**) idiogram of *M. acuminata* ssp. *zebrina* ‘Maia Oa’; (**D**) idiogram of aneuploid East African Highland banana (EAHB) clone ‘Imbogo’. Chromosomes were counterstained with DAPI (blue). Bars correspond to 5 µm. Idiograms were retrieved from Šimoníková et al. [89].

**Figure 8 plants-11-00482-f008:**
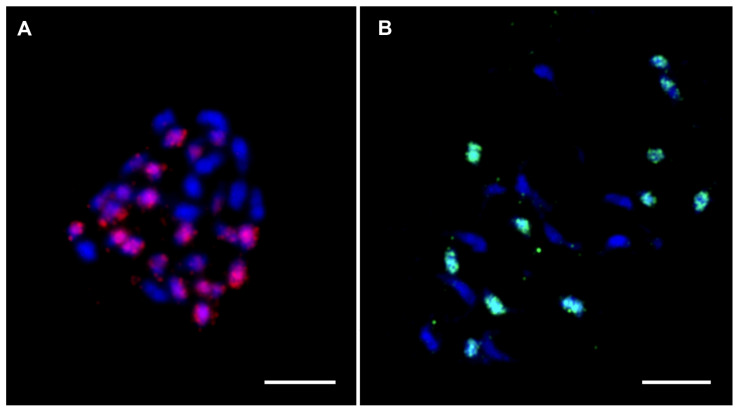
FISH on mitotic metaphase plates of two *Musa* interspecific hybrids. (**A**) Hybrid ‘Sar’ ITC1213 (2n = 3x = 32, AAT; Radka5 DNA sequence (red); (**B**) Hybrid ‘Umbubu’ ITC0854 (2n = 2x = 21, AT; Radka6 DNA sequence (green). Chromosomes were counterstained with DAPI (blue). Bars correspond to 5 µm.

## Data Availability

Not applicable.

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
