# Peer review of "Advances in the Molecular Cytogenetics of Bananas, Family Musaceae"

_plants, 2022, doi:10.3390/plants11040482_

Round 1
Reviewer 1 Report
The paper deals in great detail with cytogenetic methods in other species, connecting them with the methods and genome in bananas. Although the paper is very well written, I believe that most of the literature cited in this paper should be of more recent date.
Author Response
We thank the reviewer for positive comments. In our review, we cited most recent publications, especially in case of banana cytogenetic studies. Older citations are included mostly to outline the development and the first use of cytogenetic techniques and to support their utilization in modern, most recent cytogenetic studies.
Reviewer 2 Report
The manuscript is largely a listing of advances in banana cytogenetics. The analytical part is a bit lacking here. Conclusion "...banana, a plant species with small genome size and high chromosome number, whose interphase nuclei consists of typical heterochromatic chromocenters, can represent a model species for detailed analysis of three-dimensional organization and positioning of chromosomes during different phases of cell cycle." does not find confirmation in the text. However, there have been no reviews like this for a long time. Authors should revise the "Conclusions and future perspectives" sections.
Author Response
We thank the reviewer for positive and valuable comments and advice on how to improve the manuscript. We rewrote the "Conclusions and future perspective" section in more general manner.
Reviewer 3 Report
Even though this paper has a great potential, as currently written it is very hard to understand. I first read the manuscript thinking that it was a review paper, but I have come to realize that most (some?) are new data performed by the authors. It is actually very hard to differentiate what are the new results gathered here from the ones of previous studies.
The manuscript is not written using the MPDI template (which is fine) but also do not follow any standard template that one might use in an article. That is not necessarily wrong, but the reader needs to fully understand what was done and why – that is impossible to follow here! The aims are not stated and there are many methodological details missing throughout the manuscript. Consequently, I am not able to evaluate what was done or if any methodological flaw has occurred. For instance, results often depend on the type of instrument used, the number of particles counted, or the number of replicates used. Nothing is said about that.
Many sentences need to be text edited since they are very hard to follow.
Author Response
We thank the reviewer for valuable comments and advice how to improve our manuscript.
Our manuscript is a review paper, which includes three new results: 1) Estimation of genome size of Musella lasiocarpa and localization 45S rDNA on mitotic chromosomes of M. lasiocarpa (Part 2: Nuclear genome of bananas; Figure 1);
2) Determination of ploidy level of four Musa accessions (Part 2: Nuclear genome of bananas; Figure 2)
and 3) Identification of A- and T-genome specific chromosomes in the interspecific hybrid clone using FISH with repetitive DNA sequences (Part 4.3. Analysis of genome constitution in banana hybrid clones; Figure 8).
We add the statement "unpublished" into these three parts of our manuscript, and we also included citations on how these results were obtained.
The English was corrected by a professional editing service.
We also revised abstract.
Reviewer 4 Report
This review is well written and gives a lot of information on banana cytogenetics. The Authors write in a clear way what was done. I found only some text blocks which in my opinion are unnecessary and can be deleted:
Page 9, section 4,
This section start with some kind of an introduction into subsections 4.1, 4.2. This introduction is trivial and unnecessary, thus should be deleted. As examples:
“One of the typical chromosomal loci, being largely used as FISH probes are represented by tandem repetitive arrays of 5S and 45S rRNA genes and satellite DNA sequences which are often species-specific “
this sentence tells almost nothing. It is too general
“ Recently, short single copy sequences, mostly derived from genic regions (1.5-213 5 kb long), were also successfully used for karyotyping in some plant species, e.g. in wheat, barley or Agropyron spp..”
Why to write about wheat, barley or Agropyron instead of banana ?
Page 9 , line 224: “ Since 5S and 45S rRNA genes occur in thousands of tandemly organized copies, usually localized in separate loci in the genomes, they serve as one of the most popular cytogenetic landmarks in many plant species and are still widely used, e.g. in Aegilops spp. [51], Brassica spp. [52] or in Fragaria spp…”
Why to write about Aegilops, Brassica, Fragaria instead of banana ? Please, delete this sentence
Page 10, line 252: “ … interstitial clusters of 45S rDNA loci ..”
You should chose clusters or loci but both
Page 13, line 313 – 316 : “ Since its first successful application in cucumber [73], oligo painting FISH has become one of the most popular molecular cytogenetic techniques in plant research, enabling unambiguous identification of individual chromosomes of particular species, comparative karyotyping or validation of genome assemblies [72]. “
This review is not on oligo painting, please delete this all.
Page 17, lines 389-394: “ GISH was successfully ……. over consecutive generations [e.g. 95–97]. “
Again, this is not a review on the GISH technique – please, delete this.
Author Response
We thank the reviewer for positive and valuable comments and advice on how to improve the manuscript.
1) Page 9, section 4,
This section start with some kind of an introduction into subsections 4.1, 4.2. This introduction is trivial and unnecessary, thus should be deleted. As examples:
“One of the typical chromosomal loci, being largely used as FISH probes are represented by tandem repetitive arrays of 5S and 45S rRNA genes and satellite DNA sequences whose chromosome location is species-specific.“
this sentence tells almost nothing. It is too general
Response: We respectfully disagree and prefer to keep this sentence in the text. The reason is that these sequences have been used as informative probes for FISH.
“ Recently, short single copy sequences, mostly derived from genic regions (1.5-213 5 kb long), were also successfully used for karyotyping in some plant species, e.g. in wheat, barley or Agropyron spp..”
Why to write about wheat, barley or Agropyron instead of banana ?
Response: We have included this information to draw attention to this type of FISH probes.
Page 9 , line 224: “ Since 5S and 45S rRNA genes occur in thousands of tandemly organized copies, usually localized in separate loci in the genomes, they serve as one of the most popular cytogenetic landmarks in many plant species and are still widely used, e.g. in Aegilops spp. [51], Brassica spp. [52] or in Fragaria spp…”
Why to write about Aegilops, Brassica, Fragaria instead of banana ? Please, delete this sentence
Response: We respectfully disagree and prefer to keep this sentence in the text. The reason to include this sentence is to underline to universal use of rDNA as probes for FISH in plant molecular cytogenetic studies.
2) Page 10, line 252: “ … interstitial clusters of 45S rDNA loci ..”
You should chose clusters or loci but both
Response: We apologize for this mistake. We have modified the text accordingly.
3) Page 13, line 313 – 316 : “ Since its first successful application in cucumber [73], oligo painting FISH has become one of the most popular molecular cytogenetic techniques in plant research, enabling unambiguous identification of individual chromosomes of particular species, comparative karyotyping or validation of genome assemblies [72]. “
This review is not on oligo painting, please delete this all.
Response: We have made the suggested changes.
4) Page 17, lines 389-394: “ GISH was successfully ……. over consecutive generations [e.g. 95–97]. “
Again, this is not a review on the GISH technique – please, delete this.
Response: We respectfully disagree and prefer to keep first sentence in the text. This information was provided as a background to the discussion of the application of GISH in Musa, which follows. Hovewer, we deleted the second sentence, as suggested.
Round 2
Reviewer 3 Report
I believe the authors have addressed all my previous concerns.